# An association analysis between hypertension, dementia, and depression and the phases of pre-sarcopenia to sarcopenia: A cross-sectional analysis

Takeshi Endo[1,2☯]*, Kenju Akai[2☯], Tsunetaka Kijima[2,3☯], Shigetaka Kitahara[4‡], Takafumi Abe[2‡], Miwako Takeda[2‡], Toru Nabika[2,5‡], Shozo Yano[2,6‡], Minoru Isomura[2,7‡]

1 Division of Internal Medicine, Unnan City Hospital, Shimane, Japan, 2 Center for Community-Based Healthcare Research and Education (CoHRE), Shimane University, Shimane, Japan, 3 Department of General Medicine, Shimane University Faculty of Medicine, Shimane, Japan, 4 Kitahara Dental Clinic, Osaka, Japan, 5 Department of Functional Pathology Medicine, Shimane University Faculty of Medicine, Shimane, Japan, 6 Department of Laboratory Medicine, Shimane University Faculty of Medicine, Shimane, Japan, 7 Shimane University Faculty of Human Sciences, Shimane, Japan

☯ These authors contributed equally to this work.
‡ These authors also contributed equally to this work.
* t.endo.1211@gmail.com

**Data Availability Statement:** There are no ethical or legal restrictions on sharing a de-identified data set, which I uploaded as the additional file.

## Abstract

Sarcopenia is intricately related to aging associated diseases, such as neuropsychiatric disorders, oral status, and chronic diseases. Dementia and depression are interconnected and also related to sarcopenia. The preliminary shift from robust to sarcopenia (i.e., pre-sarcopenia) is an important albeit underdiscussed stage and is the focus of this study. Identifying factors associated with pre-sarcopenia may lead to sarcopenia prevention. To separately examine the effects of dementia and depression on pre-sarcopenia/sarcopenia, we conducted multiple analyses. This cross-sectional study used health checkup data from a rural Japanese island. The participants were aged 60 years and above, and the data included muscle mass, gait speed, handgrip strength, oral status (teeth and denture), chronic diseases (e.g., hypertension), dementia (cognitive assessment for dementia, iPad Version), and depression (self-rating depression scale). A total of 753 older adult participants were divided into the sarcopenia (n = 30), pre-sarcopenia (n = 125), and robust (n = 598) groups. An ordered logit regression analysis indicated that age and depression were positively correlated with sarcopenia, while hypertension was negatively associated with it. A multiple logistic regression analysis between the robust and pre-sarcopenia groups showed significant associations between the same three variables. Depression was associated with pre-sarcopenia, but not dementia. There was also a significant association between hypertension and pre-sarcopenia. Further research is needed to reveal whether the management of these factors can prevent sarcopenia.

**Funding:** The authors received no specific funding for this work.

**Competing interests:** The authors have declared that no competing interests exist.

## Introduction

Sarcopenia is a geriatric syndrome, characterized by age-related weakness of skeletal muscles as well as loss of muscle strength and physical function [1]. Jang et al. demonstrated the possibility that spectrums of sarcopenia may be in a continuous state of change that correlates with measures of frailty [2,3]. Exercise and nutritional therapies have been developed to prevent sarcopenia, although they are still under-discussed [4–6]. With aging, neuropsychiatric disorders (dementia and depression), oral status, history of fall, and chronic diseases are intricately linked to sarcopenia [7–14]. When analyzing the association between sarcopenia and dementia, depression may act as a confounding factor because cognitive decline due to depression may result in the person being misdiagnosed as having dementia, or in some cases, the two may be merged [15–18]. Previous research has analyzed the relationships between sarcopenia, oral frailty, chronic diseases, and dementia, but the relationship between sarcopenia and depression has so far been unexplored [19].

Older people can be classified into three stages: robust, pre-sarcopenia (i.e., decrease in muscle mass, but not in gait speed or grip strength), and sarcopenia. Pre-sarcopenia is the stage of shift from robust to sarcopenia [20]. Previous studies have shown that people who are not considered to have sarcopenia because of preserved grip strength also have an increased risk for adverse outcomes [2]. A prospective study on the association between osteoporosis and pre-sarcopenia indicated that identifying additional pre-sarcopenia–related factors could help prevent sarcopenia [21]. These issues prompted us to employ a multi-dimensional approach for the study of sarcopenia [22].

Prevention of sarcopenia is important to continue living in rural islands because of the limited number of care facilities and inpatient beds. We previously conducted observational studies in the rural areas of Shimane prefecture in Japan to analyze factors such as depression, dementia, pain, and hypertension that make it difficult for people to live there [23–27]. The issue remained to analyze the relationship between these factors and sarcopenia in rural islands. Oki-islands, which is a group of rural islands in the Shimane prefecture, consist of four islands, Dogo, Ama-town, and so on (S1 Fig). The residents of these islands were aging and displayed high rates of dementia [28]. A significant association between depression and physical activity was found in data from Dogo [29]. Another survey revealed that in Dogo, the rate of residents with hypertension is higher than the overall average of the Shimane prefecture, while the rate of medication treatment for hypertension is lower [30]. Hypertension has been reported to reduce the risk of sarcopenia [31] and is considered an important variable that needs further analysis.

The purpose of this study is to examine the variables that influence the progressing stages of sarcopenia that impact robust, pre-sarcopenia, and sarcopenia in the residents of Oki-island. We hypothesized that depression and dementia have a positive effect, while hypertension has a negative effect, on the progression stages of sarcopenia.

## Materials and methods

### Participants

We used survey data from Dogo, one of the Oki Islands in Shimane prefecture, Japan. The island has a very high aging rate at 38.4% (as reported in 2015). As of October 1, 2015, the population of Dogo was 14,608, of which 5,609 individuals were aged 60 years and above. Sampling took place in multiple health centers in Dogo, and the study population consisted of the participants of an annual health checkup conducted in June 2016. A total of 805 community-dwelling Japanese people over 60 years of age participated in this study, accounting for 14.4%

of the island's older population. The inclusion criteria for this study were as follows: (1) individuals who had been tested for sarcopenia, (2) individuals assessed for depression and dementia, and (3) individuals who consented to participate after being informed of the protocol and purpose of the current study. Excluding missing data, 753 participants were included in the analysis (Fig 1).

## Design and ethical considerations

All residents of the town of Oki Islands over the age of 40 have the right to undergo a health examination once a year with public medical insurance. This health examination covers almost all residents of this age group on the island. There are two types of health examination: group health examination at health centers and individual health examination at medical institutions. The Center for Community-Based Healthcare Research and Education (CoHRE) of Shimane University undertakes the group health examination on the island. Because the group health examination is conducted based on the individual's own initiative, there is neither verbal consent nor written consent. In addition, Japanese law does not require confirmation of dementia status. Since 2006, CoHRE has been inviting residents to participate in a cohort study (Shimane CoHRE study) to identify factors that may be causing difficulties in their lives in addition to the group health examination. This is the first analysis conducted using a cross-sectional study. The authors of this paper, who belonged to CoHRE, planned, acquired data, and created the database.

Participants in this study were all participants in the group health examination. The following procedures were used to obtain written consent for the cohort study for health screening participants: 1) To ensure that the participant (including the proxy) has time to consider participation in the study, paper information about the study was provided to the participant (including the proxy) at least once before the study was conducted. 2) On the day of the study, a new, easy-to-understand explanation was given orally according to the patient's level of understanding. 3) When consent was deemed necessary from the proxy for reasons such as cognitive decline, consent was obtained from the proxy after providing a clear explanation, while being careful not to go against the person's wishes.

The above procedures were approved by the Ethics Committee of Shimane University (#3149) and Unnan City Hospital (#20180004) as being in accordance with the Declaration of Helsinki and ethical guidelines for medical research in Japan. In this study, we used data from the 2016 Shimane CoHRE study in Oki Island town obtained through the above procedures. In this cohort study, the paper questionnaires on daily health status, cognitive function tests using iPads, and physical function tests of grip strength, body composition, and walking speed were conducted in addition to the health examination items (past medical history and other symptoms that were asked about in the interviews, height, weight, and blood pressure measurements, urinalysis, and blood sampling: liver function, lipids, and blood glucose).

## Measurement of anthropometry and skeletal muscle mass

Body height was measured using a stadiometer. Muscle mass and body weight were measured using the bioelectrical impedance analysis method with a multi-frequency segmental body composition analyzer (model MC-780A; Tanita Co., Tokyo, Japan) [32]. Body weight and muscle mass of the trunk, arms, and legs (kg) were measured, and the body-mass index (BMI) was calculated. Appendicular skeletal muscle mass was calculated as the sum of the muscle mass of the arms and legs. The muscle mass was divided by the squared height to calculate skeletal muscle index (SMI; $kg/m^2$).

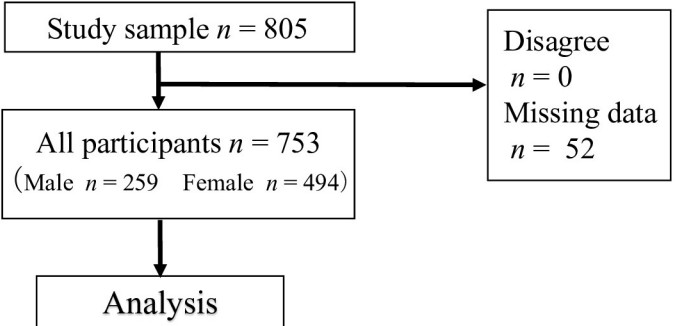

**Fig 1. Flowchart of sampling of participants.**

## Definition and classification of sarcopenia

In this study, sarcopenia was diagnosed using the criteria of the Asian Working Group for Sarcopenia (AWGS) 2014 [33]. Handgrip strength of each hand was measured twice by using a grip strength dynamometer (Takei Scientific Instruments: Niigata, Japan). Data were collected based on maximum grip strength in kilograms (kg) to the first decimal place. Usual gait speed was measured twice by trained examiners using digital stopwatches on a 5-meter course set between 3- and 8-meter marks on an 11 meter straight, flat, indoor walkway. We asked participants to walk as usual on the walkway course. Results were recorded in seconds to the first decimal place.

The other cutoff points were a grip strength of < 26 kg for men and < 18 kg for women, a usual gait speed of < 0.8 m/s, and an SMI of < 7.0 kg/m$^2$ for men and < 5.7 kg/m$^2$ for women. Sarcopenia was defined as having a low SMI in addition to low handgrip strength or low usual gait speed. Pre-sarcopenia was defined as having a low SMI with neither low usual gait speed nor low handgrip strength [34]. Robust was defined as not having a low SMI.

## Cognitive functions

**Screening for depression.** We employed the self-rating depression scale (SDS), which has high sensitivity and is suitable for screening depression [35–37]. Based on the SDS score, the following categories were used for determining the participants' state of depression: "40 points or less: no depression" "41–47 points: mild to moderate depression," and "48–55 points: moderate to severe depression" [35].

**Screening for dementia.** The Cognitive Assessment for Dementia, iPad Version (CADi2), is a dementia screening tool that can be completed using an iPad (iPad; Apple Inc., Cupertino, CA, USA). Dementia is assessed based on the number of correctly answered questions and the total response time. The questions assess 10 individual skills, such as immediate recognition and long-term memory. The CADi2 has high sensitivity and specificity, and it is based on criteria that trained persons use to diagnose dementia using neuropsychological tests (the Mini Mental State Examination or Clinical Dementia Rating Scale) [38]. The measured CADi2 score was considered as cognitive decline if the cutoff score was 6 points or less. In addition, participants who were 74 years old or younger and had a response time of 200 seconds or longer or were 75 years old or older and had a response time of 250 seconds or longer, were considered to have cognitive decline regardless of their CADi2 score. In this study, participants who were considered to have such cognitive decline were defined as the dementia group.

**Covariates.** The reported risk factors for sarcopenia are age, a history of falls, and chronic diseases [1,10,39–41]. The data for the other variables were obtained through self-report questionnaires. We inquired about age, sex (male = 0, female = 1), the number of falls within one year, and chronic diseases (hypertension, dyslipidemia, diabetes, cardio-cerebrovascular disease; no = 0, yes = 1). The presence of chronic diseases was self-reported and included if the participant had been diagnosed by a physician or received medicine.

The use of dentures and the number of remaining teeth were confirmed by a dental hygienist. These parameters are thought to be related to a decrease in masseter muscle thickness, grip strength, and gait speed [42–44].

## Statistical analysis

According to the Shapiro-Wilk normality test, all the continuous variables were not normally distributed. The characteristics of study participants in the robust, pre-sarcopenia, and sarcopenia groups were compared using Kruskal-Wallis test for continuous variables and $\chi^2$ tests for categorical variables. Mann-Whitney U test was used for the dementia and SDS scores. We calculated Spearman's correlation coefficients for continuous variables. Since the robust, pre-sarcopenia, and sarcopenia groups reflected the order of weakness, ordered logit regression analyses were conducted to determine whether there was an association between each independent variable and pre-sarcopenia and sarcopenia. History of falls was not included in the multivariate analysis because the number of falls was very low and some data were missing. The oral health parameters (remaining teeth, denture), SDS score, dementia, and chronic diseases were selected as covariables with reference to previous studies [14,19].

The following analyses were used to analyze whether these factors were associated with the shift from robust to sarcopenia: multiple logistic regression analysis between robust and pre-sarcopenia/sarcopenia, robust and pre-sarcopenia, pre-sarcopenia and sarcopenia, with the same variables as ordered logit regression analysis. According to these models, coefficients or the adjusted odds ratios (ORs) of the sarcopenia-related parameters were calculated. These multivariate analyses were available for all 753 participants.

The level of significance was set at $p < 0.05$. All data were presented as medians (interquartile range). All statistical analyses were performed using the STATA 14.2 (StataCorp, College Station, TX).

## Results

The characteristics of the 753 participants are shown in Table 1. The median age of the participants was 75.0 years, and the number of participants with pre-sarcopenia (125, 16.6%) was greater than the number with sarcopenia (30, 3.9%). Among the participants in this study, 23.6% had BMI $\geq$ 25 kg/m$^2$ and 2.5% had BMI $\geq$ 30 kg/m$^2$. In the respective order of the robust, pre-sarcopenia, and sarcopenia groups, the participants were significantly older, had a lower BMI, had fewer teeth, and were more likely to have dementia. Hypertension was significantly less common in the pre-sarcopenia group than in the robust group. Although there was no significant difference, the proportion of those using dentures and high SDS scores was higher in the robust than in the pre-sarcopenia and sarcopenia groups.

The number of participants with dementia was 49 (6.5%) and the Mann-Whitney U test did not show a significant association between dementia and SDS scores (S1 Table). The ordered logit regression analysis indicated that age and self-reported depression were positively related to sarcopenia. On the contrary, there was a significantly negative association between hypertension and sarcopenia (Table 2). Two multiple logistic regression analysis between the robust and pre-sarcopenia/sarcopenia combined groups (Table 3) and the robust

**Table 1. Baseline characteristics of the study participants according to the presence of pre-sarcopenia and sarcopenia (n = 753).**

| | Robust | | Pre-sarcopenia | | Sarcopenia | | *p*-value |
| --- | --- | --- | --- | --- | --- | --- | --- |
| | *n* = 598 | | *n* = 125 | | *n* = 30 | | |
| Gender, Female, n (%) | 392 | 65.6 | 77 | 61.6 | 25 | 83.3 | 0.08 |
| Age, median (iqr) | 74.0 | 68.0, 80.0 | 78.0 | 72.0, 83.0 | 80.0 | 77.0, 86.0 | **< 0.01** |
| Body height cm, median (iqr) | 153.6 | 148.3, 161.4 | 154.6 | 149.1, 162.3 | 148.1 | 142.5, 155.1 | **< 0.01** |
| Body weight kg, median (iqr) | 56.5 | 50.7, 64.0 | 48.0 | 44.3, 51.9 | 42.6 | 38.6, 47.9 | **< 0.01** |
| BMI kg/m2, median (iqr) | 23.6 | 21.8, 25.5 | 19.9 | 18.9, 21.1 | 20.1 | 18.4, 21.9 | **< 0.01** |
| Remaining teeth, median (iqr) | 18.0 | 6.0, 25.0 | 13.0 | 2.0, 25.0 | 12.0 | 2.0, 21.0 | 0.06 |
| Denture, n (%) | 331 | 55.4 | 79 | 63.2 | 21 | 70.0 | 0.10 |
| Hypertension, n (%) | 270 | 45.2 | 45 | 36.0 | 19 | 63.3 | **0.02** |
| Dyslipidemia, n (%) | 163 | 27.3 | 27 | 21.6 | 11 | 36.7 | 0.18 |
| Diabetes mellitus, n (%) | 59 | 9.9 | 14 | 11.2 | 7 | 23.3 | 0.06 |
| Cardio-CVD, n (%) | 81 | 13.5 | 14 | 11.2 | 4 | 13.3 | 0.54 |
| Dementia, n (%) | 30 | 5.0 | 12 | 9.6 | 7 | 23.3 | **< 0.01** |
| Fall within 1 year, mean (SD) a) | 0.3 | 1.3 | 0.2 | 0.6 | 0.4 | 0.7 | 0.37 |
| SDS, median (iqr) | 34.0 | 29.0, 39.0 | 35.0 | 31.0, 41.0 | 37.5 | 29.0, 46.0 | 0.06 |
| SDS score over 40, n (%) | 145 | 24.2 | 34 | 27.2 | 12.0 | 40.0 | 0.14 |
| Usual gait speed m/s, median (iqr) | 1.3 | 1.1, 1.4 | 1.3 | 1.1, 1.4 | 1.1 | 0.7, 1.3 | **< 0.01** |
| Hand grip strength kg, median (iqr) | 25.8 | 22.0, 34.0 | 25.4 | 21.1, 31.3 | 17.3 | 15.8, 19.0 | **< 0.01** |
| SMI kg/m2, median (iqr) | 6.8 | 6.2, 7.6 | 5.6 | 5.4, 6.4 | 5.2 | 4.9, 5.5 | **< 0.01** |

Note. ・Statistical significance of the differences between groups was determined using Kruskal-Wallis test for continuous variables and χ2 tests for categorical variables.

・Bold shows significance p < 0.05.

・OR: Odds ratios, CI: Confidence intervals.

・Abbreviations: BMI body mass index, CVD cerebrovascular disease, SDS Self-rating depression scale, SMI skeletal muscle index, iqr interquartile range, SD standard deviation.

[a]The number of data points for this variable was 735. Mean values and SD were computed instead of median because of the small sample size in the sarcopenia group.

**Table 2. Multivariate ordered logit regression analysis for robust, pre-sarcopenia, and sarcopenia (n = 753).**

| | Coef | 95% CI | *p*-value |
| --- | --- | --- | --- |
| Age | 0.09 | 0.06–0.12 | **< 0.01** |
| Sex (Female) | 0.03 | -0.38–0.43 | 0.90 |
| Remaining teeth | 0.00 | -0.02–0.03 | 0.84 |
| Denture | -0.001 | -0.53–0.53 | 0.998 |
| Hypertension | -0.47 | -0.87–0.60 | **0.02** |
| Dyslipidemia | -0.23 | -0.67–0.21 | 0.31 |
| Diabetes mellitus | 0.47 | -1.00–1.06 | 0.11 |
| Cardio-cerebrovascular disease | -0.42 | -1.00–0.16 | 0.16 |
| Dementia | 0.56 | -0.80–1.19 | 0.09 |
| Self-rating depression scale | 0.04 | 0.01–0.06 | **0.03** |

Note. Objective variables were scored as robust = 0, pre-sarcopenia = 1, sarcopenia = 2. Chronic diseases: no = 0, yes = 1.

**Table 3. Multivariate logistic regression analysis between pre-sarcopenia and sarcopenia groups combined and robust. (n = 753).**

|  | OR | 95% CI | *p*-value |
|---|---|---|---|
| Age | 1.09 | 1.06–1.13 | $<$ **0.01** |
| Sex (female) | 0.97 | 0.65–1.46 | 0.89 |
| Remaining teeth | 1.00 | 0.98–1.03 | 0.81 |
| Denture | 1.02 | 0.60–1.72 | 0.95 |
| Hypertension | 0.60 | 0.40–0.90 | **0.01** |
| Dyslipidemia | 0.79 | 0.51–1.24 | 0.31 |
| Diabetes mellitus | 1.50 | 0.83–2.70 | 0.18 |
| Cardio-cerebrovascular disease | 0.66 | 0.37–1.20 | 0.17 |
| Dementia | 1.65 | 0.86–3.15 | 0.13 |
| Self-rating depression scale | 1.03 | 1.01–1.06 | **0.01** |

Note. Objective variables were scored as robust = 0, pre-sarcopenia and sarcopenia combined group = 1. Chronic diseases: no = 0, yes = 1.

and pre-sarcopenia groups (Table 4) showed that the three variables were similarly significantly associated: age, self-reported depression, and hypertension were related to sarcopenia. Conversely, in the multiple logistic regression analysis between the pre-sarcopenia and robust groups, the only factor significantly associated was sex (i.e., female) (Table 5). The variance inflation factors of the variables used in the multivariate analysis conducted in this study were less than 10.

S2 Table shows the Spearman's correlation coefficients. The BMI was not included as a variable in the multivariate analysis because of its high correlation with SMI used in the diagnostic criteria for sarcopenia. The variance inflation factors of the variables used in the multivariate analysis conducted in this study were less than 10. S3 Table shows the number of participants who fell below the diagnostic criteria for sarcopenia was divided by sex and sub-category (pre-sarcopenia and sarcopenia).

## Discussion

According to the robust, pre-sarcopenia, and sarcopenia groups, results showed that participants were older in age, had a lower BMI, fewer number of teeth, and a higher SDS score, and

**Table 4. Multivariate logistic regression analysis between robust and pre-sarcopenia (n = 723).**

|  | OR | 95% CI | *p*-value |
|---|---|---|---|
| Age | 1.08 | 1.05–1.12 | $<$ **0.01** |
| Sex (female) | 0.79 | 0.51–1.22 | 0.29 |
| Remaining teeth | 1.00 | 0.97–1.03 | 0.91 |
| Denture | 0.97 | 0.54–1.73 | 0.91 |
| Hypertension | 0.51 | 0.33–0.80 | $<$ **0.01** |
| Dyslipidemia | 0.74 | 0.45–1.21 | 0.23 |
| Diabetes mellitus | 1.27 | 0.65–2.47 | 0.49 |
| Cardio-cerebrovascular disease | 0.68 | 0.36–1.29 | 0.24 |
| Dementia | 1.25 | 0.59–2.63 | 0.56 |
| Self-rating depression scale | 1.03 | 1.00–1.06 | **0.04** |

Note. Objective variables were scored as robust = 0, pre-sarcopenia = 1. Chronic diseases: no = 0, yes = 1.

**Table 5. Multivariate logistic regression analysis between pre-sarcopenia (0) and sarcopenia (1) (n = 155).**

|  | OR | 95% CI | *p*-value |
|---|---|---|---|
| Age | 1.06 | 0.97–1.15 | 0.18 |
| Sex (Female) | 5.66 | 1.50–21.4 | **0.01** |
| Remaining teeth | 0.98 | 0.93–1.05 | 0.63 |
| Denture | 0.96 | 0.27–3.43 | 0.95 |
| Hypertension | 1.76 | 0.68–4.59 | 0.25 |
| Dyslipidemia | 1.28 | 0.48–3.44 | 0.62 |
| Diabetes mellitus | 3.24 | 0.91–11.5 | 0.07 |
| Cardio-cerebrovascular disease | 1.19 | 0.30–4.70 | 0.81 |
| Dementia | 2.79 | 0.82–9.48 | 0.10 |
| Self-rating depression scale | 1.05 | 0.99–1.11 | 0.09 |

Note. Objective variables were scored as pre-sarcopenia = 0, sarcopenia = 1. Chronic diseases: no = 0, yes = 1.

a higher proportion had dementia. These results indicated that a steady decline in individuals' mental and physical status, led to the development of sarcopenia; moreover, this finding is consistent with a previous study, indicating that the shift from robust to sarcopenia is a continuous state of change that correlates with frailty [2].

We found multiple factors associated with the progression from robust to pre-sarcopenia and sarcopenia. In the progression of stages from robust to pre-sarcopenia, in addition to age, depression and hypertension were significantly associated with the participants' status.

The high depressive tendency in our study might be the reason why there was a significant association between self-reported depression and sarcopenia. A report showed that increased depression severity was associated with sarcopenia [45]. The relationship between depression and sarcopenia remains under discussion. In the Korean study that did not find a significant association between sarcopenia and depression, 16.6% of the sarcopenia participants and 14.4% of the non-sarcopenia participants had depressive tendencies [46]. Others have reported the opposite [14,47,48]. Hsu et al. reported that the rate of depressed participants was 29.8% in the sarcopenia group and 14.3% in the non-sarcopenia group and depression was significantly associated with sarcopenia [14]. In our study, the proportion of participants judged as depressed (a score over 40 points on the SDS) was even higher. The relationship between depression and sarcopenia can be justified by the commonly associated factors such as inactivity [49] and chronic inflammation [50,51].

In the present study, there was no significant association between dementia and sarcopenia. The possible reason why sarcopenia and dementia were not significantly related was the small number of dementia and sarcopenia participants. In the present study, there was no significant association between dementia and sarcopenia. The sarcopenia rate in our study was 3.9%, lower than in a previous study which reported a rate of approximately 10.0% [52,53]. This may be because the participants in our study were active enough to participate in a voluntary health checkup, whereas bedridden and institutionalized residents could not participate in the checkups. The dementia rate in our study was 6.5%, which was smaller than that reported by a nearby island's study showing 16.4% dementia in the general population of Ama-town [28]. Hsu et al. (2014) indicated that both dementia and depression were significantly associated with sarcopenia. In their study, the rate of participants with sarcopenia was 30.9% and of those with dementia was 38.0% [14]. These rates were clearly different from those in our study. Other studies analyzing the relationship between sarcopenia and dementia did not include depression as a variable; hence, they could not be compared with our study [54–56].

Depression did not have a confounding effect on the association between dementia and sarcopenia. Firstly in our study, there was no significant association between the SDS score and dementia, which indicated that depressive tendencies did not significantly influence the CADi test to determine dementia, although a previous study showed that depression affects cognitive function [17]. Secondly, we conducted the ordered logit regression analyses including both dementia and depression.

In this study, hypertension was significantly less common in pre-sarcopenia patients than in robust patients. This was consistent with a previous study [31]. In general, nutritional therapy and exercise therapy are recommended for the management of hypertension [57], and these two therapies have been reported to be effective in preventing sarcopenia [4,5]. However, the relationship between hypertension and prevention of sarcopenia requires further research.

In this study, there were significantly more females in the sarcopenia group than in the pre-sarcopenia group. The two groups are defined by the difference in muscle strength in which pre-sarcopenia has preserved muscle strength while sarcopenia has decreased muscle strength. According to the AWGS 2014 criteria for sarcopenia, this muscle strength is defined as SMI (<5.7 kg/m2 for female and <7.0 kg/m2 for male) and grip strength (<18 kg for female and <26 kg for male), with a lower cutoff point for women. In the sarcopenia group of this study, it was clear that there were more females with grip weakness than male (S3 Table). Further studies are needed to clarify the causal relationship between sex and this muscle weakness.

## Limitations

There are several limitations in this study. First, this was a cross-sectional study; as such, a prospective study will be required to investigate the causal relationships between sarcopenia and its related factors. The second limitation is that participation in this cohort study was voluntary, and a random selection of Dogo residents was not used, therefore, selective bias may have occurred. Third, this study was conducted in 2016, and the speed of the 5 m gait was measured according to the AWGS 2014 criteria and not the AWGS 2019 criteria using the 6 m gait. Our future task is to conduct a sarcopenia study using the AWGS 2019 criteria.

## Conclusion

Depression was positively associated with the sarcopenia group compared to the robust one, while hypertension was negatively associated. Dementia, however, had no significant effect. The depression scale of the rural island residents needs to be followed up because it can be an associated factor in the future progression of sarcopenia.

## Supporting information

**S1 Fig. Map showing the location of Dogo, which is one of the Oki Islands.** Published with permission from Oki Islands UNESCO Global Geopark Promotion Committee. Available from: http://www.oki-geopark.jp/en/features/ (accessed July 22, 2020).
(TIF)

**S1 Table. Mann-Whitney U test for the self-rating depression scale according to the presence of dementia (n = 753).**
(TIFF)

**S2 Table. Spearman's rank correlations between the participants' characteristics and sarcopenia-related factors.** * $p < 0.10$,** $p < 0.05$, *** $p < 0.01$.
(TIFF)

**S3 Table. The number of participants who fell below the diagnostic criteria for sarcopenia was divided by sex and sub-category (pre-sarcopenia and sarcopenia).**
(TIFF)

**S1 Dataset.**
(XLSX)

## Author Contributions

**Conceptualization:** Kenju Akai, Tsunetaka Kijima, Shigetaka Kitahara.

**Data curation:** Miwako Takeda.

**Formal analysis:** Takafumi Abe.

**Methodology:** Shozo Yano.

**Supervision:** Toru Nabika, Minoru Isomura.

**Writing – original draft:** Takeshi Endo.

**Writing – review & editing:** Kenju Akai, Tsunetaka Kijima.

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
