## [Decision Letter · Decision Letter 0]

14 Dec 2020

PONE-D-20-34154

The transition phase of pre-sarcopenia to sarcopenia association analysis between hypertension, dementia, and depression: A cross-sectional study

PLOS ONE

Dear Dr. Endo,

Thank you for submitting your manuscript to PLOS ONE. After careful consideration, we feel that it has merit but does not fully meet PLOS ONE’s publication criteria as it currently stands. Therefore, we invite you to submit a revised version of the manuscript that addresses the points raised during the review process.

The manuscript by Endo et al. does not reach to an enough level for the acceptance in the Journal.

See the three Reviewers' comments carefully and respond them appropriately.

We look forward to receiving your revised manuscript.

Kind regards,

Masaki Mogi

Academic Editor

PLOS ONE

Journal Requirements:

2.Thank you for stating the following financial disclosure:

 [The funders had no role in study design, data collection and analysis, decision to publish, or preparation of the manuscript.].

3.We note that you have indicated that data from this study are available upon request. PLOS only allows data to be available upon request if there are legal or ethical restrictions on sharing data publicly. For information on unacceptable data access restrictions, please see http://journals.plos.org/plosone/s/data-availability#loc-unacceptable-data-access-restrictions.

4.We note that [Figure(s) 1] in your submission contain map images which may be copyrighted. All PLOS content is published under the Creative Commons Attribution License (CC BY 4.0), which means that the manuscript, images, and Supporting Information files will be freely available online, and any third party is permitted to access, download, copy, distribute, and use these materials in any way, even commercially, with proper attribution. For these reasons, we cannot publish previously copyrighted maps or satellite images created using proprietary data, such as Google software (Google Maps, Street View, and Earth). For more information, see our copyright guidelines: http://journals.plos.org/plosone/s/licenses-and-copyright.

1.    You may seek permission from the original copyright holder of Figure(s) [1] to publish the content specifically under the CC BY 4.0 license. 

Reviewers' comments:

Reviewer's Responses to Questions

**Comments to the Author**

1. Is the manuscript technically sound, and do the data support the conclusions?

Reviewer #1: Yes

Reviewer #2: No

Reviewer #3: Yes

2. Has the statistical analysis been performed appropriately and rigorously? 

Reviewer #1: Yes

Reviewer #2: No

Reviewer #3: Yes

3. Have the authors made all data underlying the findings in their manuscript fully available?

Reviewer #1: Yes

Reviewer #2: No

Reviewer #3: Yes

4. Is the manuscript presented in an intelligible fashion and written in standard English?

Reviewer #1: Yes

Reviewer #2: No

Reviewer #3: Yes

5. Review Comments to the Author

Reviewer #1: Thank you for your invitation. This is a paper addressing the association between sarcopenia, pre-sarcopenia, and depression in a rural population of older adults. Some important issues need to be considered.

1. Introduction: 1. The authors stated that "sarcopenia is a phenotype of frailty". This statement is not based on evidence. It was just a consideration from some researchers. Some authors argued that frailty and sarcopenia were "two sides of one coin." Please try to use the widely agreement of the definition of sarcopenia, such as EWGSOP2 or AWGS.

2. Introduction: 'Sarcopenia can be classified into three stages: robust, pre-sarcopenia, and sarcopenia'. This statement is not true. Robust is not a stage of sarcopenia. Sarcopenia can be classified into three stages: pre-sarcopenia, sarcopenia, and severe sarcopenia.

3. Figure 1 is not the main results of this paper. It should serve as a supplementary figure, if necessary.

4. This a secondary analysis of a cross-sectional study. This point shouild be clearly stated.

5. We need more details about the handgrip strength measurement and walking speed measurement.

6. The authors defined sarcopenia based on the AWGS 2014. Why didn't they use the AWGS 2019? Would the results significantly change if the AWGS 2019 definition of sarcopenia was applied? Perhaps, the authors can also use the AWGS 2019 criteria and reanalysis their data to perform a sensitivity analysis of their results in order to test the robustness of their conclusion.

7. "Chronic diseases (hypertension, dyslipidemia, diabetes, cardio- 169

cerebrovascular disease; no = 0, yes = 1)" What about other important diseases such as COPD, liver disease, and CKD?

8. Why did the authors not define severe sarcopenia since they have the data on ASMI, handgrip strength, and walking speed?

9. Did the authors consider the multicolinearity in their models?

10. The authors reported that “all the continuous variables were not normally distributed” . However, they reported mean and SD for these variables in Table 1.

11. The prevalence of sarcopenia is extremely low (3.9%) compared to the data in the literature. Is there any explanation for these results? Maybe there are selection bias as the participants were recruited from group health examination. As a result, the generalizability of this paper would be weakened.

Reviewer #2: I found the study titled " The transition phase of pre-sarcopenia to sarcopenia association analysis between hypertension, dementia, and depression: A cross-sectional study" deals an interesting topic on spectrum of sarcopenia in community-dwelling older adults. However, I noted that there are tremendous rooms for improvement for the manuscript to provide enough explanation that the data deserves, and to add some clinical knowledge in the field of geriatrics.

Major points>

1) Throughout the manuscript: By using the term ‘transition’, authors may mislead readers on the nature of the study, since this cross-sectional study only compares clinical factors across non-sarcopenic, pre-sarcopenic, and sarcopenic participants in this community-based study while the term transition usually denotes some longitudinal dynamic changes of a condition.

2) Introduction: This part should be improved to provide rational backgrounds and needs to support the conceptualization of the study hypothesis. Irrelevant remarks in the introduction part (e.g., line 57-58, Cognitive decline due to…) should be removed. In the last part, study hypothesis should be more clearly addressed. Also, on top of previous researches that authors remarked in the last part of the introduction, I hope to see what the current study adds up as a meaningful scientific literature.

3) Analysis/results: While authors used logistic regression analysis, I think factors associated with sarcopenia spectrum can be more clearly assessed using ANCOVA, with no-sarcopenia, presarcopenia, and sarcopenia as independent variables. Maybe as a sensitive analysis. As a recent paper suggested (Jang et al, JCSM 2020 Apr; 11(2): 497–504), spectrums of sarcopenia might be incremental and somehow continuous, highly correlate with frailty spectrum. Therefore, performing multiple comparisons between 3 conditions of no sarcopenia, pre-sarcopenia, and sarcopenia might be less meaningful.

4) Discussion, Line 288: In the current study design, mechanism cannot be drawn. These remarks seem to be rather speculative.

4) Discussion, Line 307: Irrelevant (or less relevant) remarks for current study and analysis.

5) Conclusion: Conclusion should concisely summarize the main hypothesis and corresponding study results, and interpretations from authors.

Minor points>

1) Co-first authors are not marked in the author list

2) I recommend authors rather not to use the term elderly in the scientific literature, (See Vaughan et al, JAGS, 2019 67:211-217), and replace it with case-specific terms such as older people, older participants, individuals etc. While authors are using the term ‘subject’, I also noted that this word can be replaced with ‘participants’ in the present study is an observational study. In choosing words describing sex/gender, I recommend to have some unity, for example sex/male/female vs. gender/men/women

3) Page 8 Line 131, Is the term physique a commonly used word in scientific literature?

4) Table 1, n=753 is everywhere, except for fall item (n=735). I recommend authors to provide total n (753) in general), and provide n of fall item in the note with appropriate quotation marks for the journal. P values should be provided with 3 digits below decimal points. I also opposite in separating table into 2 vertical sections of mean+-sd and n(%), but this might be up to the journal’s policy.

Reviewer #3: This study of a rural Japanese island is very interesting and worthy of publication. The authors suggested that aging, depression, and hypertension were associated with sarcopenia in rural islands. However, there are several major concerns with the manuscript that must be addressed.

Major comments:

1. The authors described traditional risk factors for sarcopenia and the current state of the rural island in the introduction. However, the purpose of the study was unclear, so please include this at the end of the introduction.

2. The presence or absence of obesity (≥25 or 30 kg/m2) is also an important independent factor for sarcopenia; however, why did you mention hypertension, diabetes, and cerebrovascular disease without including obesity?

3. The comparison of the three groups was performed using the Kruskal-Wallis test or χ2 test, but no ad hoc analysis was performed, such as multiple comparison or residual analysis.

4. The prevalence of sarcopenia in this study was 3.9%, which is very low compared with those reported in other studies. It is necessary to explain the reason and characteristics of the low prevalence of sarcopenia, including the study of rural islands.

5. There were sex-based differences in this study. Please clarify why you did not analyze by sex.

Minor comments:

1. It is recommended that the island description and figure 1 provided in the introduction are presented in the participants section of Materials and Methods.

2. The current diagnosis of sarcopenia generally uses AWGS2019; however, it is important to explain why AWGS was used.

3. Please add the definition or cutoff for dementia diagnosis by CADi2 in the “screening for dementia” section.

4. The non-normal distribution data in Table 1 should be presented as median and interquartile range.

5. The following research showed that hypertension was a protective factor for sarcopenia. Hence, I recommend that you refer to it as supporting evidence:

Kurose S et al, Sci Rep 10:19129, 2020.

6. PLOS authors have the option to publish the peer review history of their article (what does this mean?). If published, this will include your full peer review and any attached files.

Reviewer #1: No

Reviewer #2: No

Reviewer #3: No

---

## [Author Response · Author response to Decision Letter 0]

24 Apr 2021

Response to Reviewer 1

Thank you for pointing out the most fundamental part of the definition and classification of sarcopenia. Please also note that we have included a description of why we did not follow the AWGS 2019 criteria.

1. Introduction: 1. The authors stated that "sarcopenia is a phenotype of frailty". This statement is not based on evidence. It was just a consideration from some researchers. Some authors argued that frailty and sarcopenia were "two sides of one coin." Please try to use the widely agreement of the definition of sarcopenia, such as EWGSOP2 or AWGS.

Reply: As you pointed out, the description seemed to confuse sarcopenia with frailty, so we changed it to something widely agreed upon.

Revised: Page 4, line 51-54

2. Introduction: 'Sarcopenia can be classified into three stages: robust, pre-sarcopenia, and sarcopenia'. This statement is not true. Robust is not a stage of sarcopenia. Sarcopenia can be classified into three stages: pre-sarcopenia, sarcopenia, and severe sarcopenia.

Reply: Thank you for pointing out this error. We had to differentiate between older people’s medical condition and not sarcopenia. We did not include severe sarcopenia because the purpose of this study was to analyze the characteristics of pre-sarcopenia. Furthermore, the study was conducted in 2016, and we followed the criteria of AGWS 2014 and not AGWS 2019, which incorporated severe sarcopenia. Please see the response to comments 6 and 8 for a more detailed explanation. We have corrected it as follows.

Revised: Page 4, line 63-65

3. Figure 1 is not the main results of this paper. It should serve as a supplementary figure, if necessary.

Reply: I followed your instructions and changed it to a supplementary file.

4. This a secondary analysis of a cross-sectional study. This point should be clearly stated.

Reply: This is the first analysis conducted by a cross-sectional study. The authors of this paper, who belonged to CoHRE, planned, acquired data, and created the database. This part was not written properly, so I added it to the Method section.

Revised: Page 7-8, line 116-1185. We need more details about the handgrip strength measurement and walking speed measurement.

Reply: Thank you for pointing this out. I added a note about the measurement method and its definition of decline. 

Revised: Page 9-10, Line 148-154 <Method>

6. The authors defined sarcopenia based on the AWGS 2014. Why didn't they use the AWGS 2019? Would the results significantly change if the AWGS 2019 definition of sarcopenia was applied? Perhaps, the authors can also use the AWGS 2019 criteria and reanalysis their data to perform a sensitivity analysis of their results in order to test the robustness of their conclusion.

Reply: We believe that in remote areas with few medical resources, prevention of sarcopenia is an important issue, and this study focused on extracting the characteristics of pre-sarcopenia. Since the purpose of this study was not properly communicated, I have included the purpose in the last sentence of the introduction so that the purpose is clear again. 

The sarcopenia study you mentioned, using the AWGS 2019 criteria, is our next research study. CoHRE is still conducting annual surveys in the same way, and a paper on sarcopenia using the AWGS2019 criteria is being prepared using data from Oki Island.

7. "Chronic diseases (hypertension, dyslipidemia, diabetes, cardio-cerebrovascular disease; no = 0, yes = 1)" What about other important diseases such as COPD, liver disease, and CKD?

Reply: We did not obtain the data that you mentioned at the time of our 2016 survey. We are currently acquiring data on liver diseases and will consider acquiring data on COPD and CKD in the future.

8. Why did the authors not define severe sarcopenia since they have the data on ASMI, handgrip strength, and walking speed?

Reply:

This study was conducted in 2016, and the speed of the 5m gait was measured according to the AWGS 2014 criteria, not the 6m gait used in AWGS 2019. Other methods of case selection such as lower leg circumference and SARC-F recommended by AWGS 2019 were not assessed. Therefore, it is difficult to reanalyze the data using the criteria of AWGS 2019 because the inclusion criteria was different. As a trial, we tried to diagnose severe sarcopenia using the AWGS 2019 criteria using SMI, walking speed, and hand grip strength, but the number of participants was as low as nine. The participants were community-dwelling health screening participants, who had a higher percentage of pre-sarcopenia than sarcopenia. As we noted in reply to comment 6, a sarcopenia study using the AWGS 2019 criteria is our future research.

Revised: Page 6, Line 95-96

9. Did the authors consider the multicolinearity in their models?

Reply: First, the correlation coefficients between each variable were calculated, and adjustments were made so that variables that were strongly correlated, such as BMI and SMI, were not entered simultaneously. Further, both the ordered logistic regression and the variance inflation factor of the variables found in the multiple logistic regression analysis were less than 10. This was added to the results.

Revised: Page 16, Line 246-247 <Result>

10. The authors reported that “all the continuous variables were not normally distributed” . However, they reported mean and SD for these variables in Table 1.

Reply: I created a new table 1 with changes to median and inter quartile range.

11. The prevalence of sarcopenia is extremely low (3.9%) compared to the data in the literature. Is there any explanation for these results? Maybe there are selection bias as the participants were recruited from group health examination. As a result, the generalizability of this paper would be weakened.

Reply: As mentioned in Page19, line 278-281, this study was based on health examination data of relatively active older people who were able to come voluntarily to the study site, and the incidence of sarcopenia was low. The main purpose of this study was to pick up relevant factors in participants with pre-sarcopenia, and we believe that the design was able to fulfill that purpose. However, as you pointed out, the findings of this study do not apply to populations with a high prevalence of sarcopenia, such as community-dwelling older people, including those who are bedridden in home care or institutionalized patients. This was not explained well enough, so I added this information in the discussion section.

Revised:　Page 22, Line 311-313 <Discussion>

Response to Reviewer 2

We greatly appreciate your comments. I learned that sarcopenia is some kind of continuous state change and have revised our manuscript accordingly. 

Major points>

1) Throughout the manuscript: By using the term ‘transition’, authors may mislead readers on the nature of the study, since this cross-sectional study only compares clinical factors across non-sarcopenic, pre-sarcopenic, and sarcopenic participants in this community-based study while the term transition usually denotes some longitudinal dynamic changes of a condition.

Reply: As you pointed out, we realized that using "transition" in our research results misleads people into thinking that it is a longitudinal study. We deleted the word “transition” and changed it to "shift”.

Revised: Page 2, line 27, page 4, line 65, page 6, line 86, page 13, line 207 and page 21, line 289

2) Introduction: This part should be improved to provide rational backgrounds and needs to support the conceptualization of the study hypothesis. Irrelevant remarks in the introduction part (e.g., line 57-58, Cognitive decline due to…) should be removed. In the last part, study hypothesis should be more clearly addressed. 

Reply: The parts you have highlighted were not clearly related to the main discussion; hence, I have deleted the text and made revisions accordingly.

Revised: Page 4, Line 57-60

2) Also, on top of previous researches that authors remarked in the last part of the introduction, I hope to see what the current study adds up as a meaningful scientific literature.

Reply: We believe that prevention of sarcopenia in remote areas where medical resources are scarce is important, and our main focus was to extract the characteristics of pre-sarcopenia. Since we did not explain this purpose properly, we stated it in the last sentence of the introduction to clearly convey the purpose.

Revised: Page 6, Line 84-88

3) Analysis/results: While authors used logistic regression analysis, I think factors associated with sarcopenia spectrum can be more clearly assessed using ANCOVA, with no-sarcopenia, presarcopenia, and sarcopenia as independent variables. Maybe as a sensitive analysis. As a recent paper suggested (Jang et al, JCSM 2020 Apr; 11(2): 497–504), spectrums of sarcopenia might be incremental and somehow continuous, highly correlate with frailty spectrum. Therefore, performing multiple comparisons between 3 conditions of no sarcopenia, pre-sarcopenia, and sarcopenia might be less meaningful.

Reply: Thank you for suggesting a better analysis. The purpose of this study is to analyze the process of transition to sarcopenia.

We are in complete agreement that spectrum of sarcopenia is some kind of a continuous. Therefore, we also believe that sarcopenia is a gradual and sequential transition, such as robust, pre-sarcopenia, and sarcopenia, and we conducted ordered logistic regression, a logit analysis that can track the sequential transition. As you pointed out, ANCOVA may be more appropriate if these three are considered continuous. ANCOVA with robust, pre-sarcopenia, and sarcopenia assigned as dependent variables (0, 1, and 2) showed a significant association between age and diabetes mellitus (additional file). Thank you for your guidance on other analysis methods. However, we would like to leave this as a future issue because looking at the correlation between sarcopenia and frailty spectrum with this kind of method is far from the main purpose of this study.

In our introduction and discussion sections, I have added the findings that sarcopenia is some kind of continuous state that correlates with frailty index (Jang et al, JCSM 2020 Apr; 11(2): 497–504). I also learned that the frailty index used here has been studied in the Asian region for a long time and is an index that can be applied to Japanese people without problems (Jung et al, PLoS One. 2014 Feb 4;9(2):e87958). I have also added this finding to the text.

This study is consistent with the purpose of our research, which showed that people with pre-sarcopenia, who are not considered to have sarcopenia because their grip strength is preserved, also have a risk for adverse outcomes. Also, the purpose of comparing these three states and the hypothesis is more clearly stated in the last paragraph of the introduction. Thank you for your suggestion.

Revised: Page 4, Line 52-54 and 65-66<Introduction>

Page 5, Line 66-69 <introduction>

Previous studies have shown that people who are not considered to have sarcopenia because of preserved grip strength also have an increased risk for adverse outcomes .

Page 6, Line 84-88 <Introduction>

The purpose of this study is to examine the variables that influence the process of transition to sarcopenia in the residents of Oki-island. We hypothesized that depression and dementia would affect this shift and conducted ordered logit analysis between these factors, in addition to oral status and hypertension, on pre-sarcopenia and sarcopenia. 

Page 21, Line 287-290

These results indicate when there is a steady decline in individuals’ mental and physical status, they develop sarcopenia, and this finding is consistent with a previous study, indicating that the shift from robust to sarcopenia is a continuous state of change that correlates with frailty .

4) Discussion, Line 288: In the current study design, mechanism cannot be drawn. These remarks seem to be rather speculative.

Reply: You pointed out the part where you speculate why hypertension was less in pre-sarcopenia patients. We have added a previous study on the relationship between hypertension and pre-sarcopenia and have stated that our results are consistent with them and made some assumptions about the mechanisms involved.

Revised: Page 23, Line 327-332<Discussion>

5) Discussion, Line 307: Irrelevant (or less relevant) remarks for current study and analysis.

Reply: We have confirmed that the content was indeed irrelevant to the text. Therefore, I have deleted the text.

6) Conclusion: Conclusion should concisely summarize the main hypothesis and corresponding study results, and interpretations from authors.

Reply: I clarified the description of the hypothesis in the introduction, and based on that, I revised the conclusion per your suggestion.

Revised: Page 25, Line 351-353 <Conclusion>

Minor points>

1) Co-first authors are not marked in the author list

Reply: The co-first authors are now marked on the title page.

Revised: Page 1, Line 5-6

2) I recommend authors rather not to use the term elderly in the scientific literature, (See Vaughan et al, JAGS, 2019 67:211-217), and replace it with case-specific terms such as older people, older participants, individuals etc. While authors are using the term ‘subject’, I also noted that this word can be replaced with ‘participants’ in the present study is an observational study. In choosing words describing sex/gender, I recommend to have some unity, for example sex/male/female vs. gender/men/women

Reply: We have unified the terminology as you suggested.

Revised: Throughout the entire manuscript.

3) Page 8 Line 131, Is the term physique a commonly used word in scientific literature?

Reply: The term "physique" has been replaced by "anthropometry" because of a misinterpretation of the terminology.

Revised: Page 9, Line 138 <Method>

4) Table 1, n=753 is everywhere, except for fall item (n=735). I recommend authors to provide total n (753) in general), and provide n of fall item in the note with appropriate quotation marks for the journal. P values should be provided with 3 digits below decimal points. I also opposite in separating table into 2 vertical sections of mean+-sd and n(%), but this might be up to the journal’s policy.

Reply: We have revised Table 1 accordingly.

Revised: Page 14, Line 227

Response to Reviewer 3

We greatly appreciate your specific guidance on what needs to improve in terms of BMI numbers, incidence of sarcopenia, and AGWS criteria. 

Major comments:

1. The authors described traditional risk factors for sarcopenia and the current state of the rural island in the introduction. However, the purpose of the study was unclear, so please include this at the end of the introduction.

Reply: As you pointed out, the part describing the purpose of the research was unclear; hence, I have added it to the last paragraph of the introduction.

Revised: Page 4, Line 52-54 and 65-66<Introduction>

2. The presence or absence of obesity (≥25 or 30 kg/m2) is also an important independent factor for sarcopenia; however, why did you mention hypertension, diabetes, and cerebrovascular disease without including obesity?

Reply: As you pointed out, BMI is an important index that is also related to muscle mass and mortality [Abramowitz MK, 2018;13(4):e0194697. Epub 2018/04/12], and obesity has been reported to be a risk factor for sarcopenia [Batsis JA, Nat Rev Endocrinol. 2018;14(9):513-37]. BMI was not included as a covariate because skeletal muscle index, one of the diagnostic criteria for sarcopenia, had a very high correlation with BMI. Including BMI in the ordered logistic regression with Robust/Pre-sarcopenia/Sarcopenia as the dependent variable would result in high multicollinearity. However, obesity is a very important index for sarcopenia, and we have added it to the results.

Revised: Page 14, Line 219-220<Result>

Among the participants in this study, 23.6% had BMI ≥ 25 kg/m2 and 2.5% had BMI ≥ 30 kg/m2.

3. The comparison of the three groups was performed using the Kruskal-Wallis test or χ2 test, but no ad hoc analysis was performed, such as multiple comparison or residual analysis.

Reply: The purpose of our study is to show a stepwise trend and not to look at the difference between two groups in an ad hoc multiple comparison. For this reason, we used ordered logistic regression as our method. I explicitly added the reason for using this method in the analysis section.

Reviewer 2 also suggested that we use ANCOVA, and we explained accordingly.

4. The prevalence of sarcopenia in this study was 3.9%, which is very low compared with those reported in other studies. It is necessary to explain the reason and characteristics of the low prevalence of sarcopenia, including the study of rural islands.

Reply: We thought that the lower prevalence of sarcopenia in this study than in previous studies was due to the fact that the participants were relatively active older people who were able to voluntarily come to the study site. This was also mentioned in Page 22, Line 311-313, but there was no "comparison with sarcopenia prevalence in remote island" as you pointed out. In addition, the fact that the participants did not include bedridden or institutionalized residents may have contributed to this low rate, so we have made the following changes.

Revised: Page 22, Line 311-313

5. There were sex-based differences in this study. Please clarify why you did not analyze by sex.

Reply: As you pointed out, sex was a significantly related variable between the pre-sarcopenia and sarcopenia groups (Table 3.3). However, in the characteristics table (Table 1) and the ordered logistic regression table (Table 2), and multiple logistic regression (Table 3.1, 3.2) sex was not a significantly associated factor. Since the main purpose of this study was to extract the characteristics of pre-sarcopenia and other stages, we did not conduct the analysis with sex as the main axis. The difference between pre-sarcopenia and sarcopenia is not in muscle mass but in muscle strength, that’s why sex may be related to this part. I also made a comparison table of SMI, hand grip strength, and usual gait speed of the two groups and added it to the results as an additional file (S3 table). The reason for the association between women and sarcopenia is that there were more women with sarcopenia who had low hand grip strength. Thank you for presenting a new perspective. I have added the following to the text.

Revised: Page 24, Line 333-341<Discussion>

Minor comments:

1. It is recommended that the island description and figure 1 provided in the introduction are presented in the participants section of Materials and Methods.

Reply: I changed the location of figure 1 as you instructed. Also, as pointed out by another reviewer, I made this figure an additional file because this figure is not what we have clarified in this study.

2. The current diagnosis of sarcopenia generally uses AWGS2019; however, it is important to explain why AWGS was used.

Reply: We believe that in remote areas with few medical resources prevention of sarcopenia is an important issue, and this study focused on extracting the characteristics of pre-sarcopenia. Since the purpose of this study was not properly communicated, I have included it in the last sentence of the introduction so that the purpose is clear again. Also, this study was conducted in 2016, and the speed of the 5m walk was measured according to the AWGS 2014 criteria, not the 6m walk used in AWGS 2019. Other methods of case selection such as lower leg circumference and SARC-F recommended by AWGS 2019 were not assessed. Therefore, it is difficult to reanalyze the data using the criteria of AWGS 2019 because the inclusion criteria was different. As a trial, we tried to diagnose severe sarcopenia using the AWGS 2019 criteria using SMI, walking speed, and hand grip strength, but the number of participants was as low as nine. The participants were community-dwelling health screening participants, who had a higher percentage of pre-sarcopenia than sarcopenia. CoHRE is still conducting annual surveys in the same way, and a paper on sarcopenia using the AWGS 2019 criteria is being prepared using data from Oki Island.

Revised: Page 6, Line 95-96　<Introduction>

3. Please add the definition or cutoff for dementia diagnosis by CADi2 in the “screening for dementia” section.

Reply: As you pointed out, the cut off was unclear. This cognitive function test is somewhat complicated, as the test takes into account not only the score but also the response time. I have added the information to the Method in the text.

Revised: Page 11, Line 175-181, <Method>

4. The non-normal distribution data in Table 1 should be presented as median and interquartile range.

5. The following research showed that hypertension was a protective factor for sarcopenia. Hence, I recommend that you refer to it as supporting evidence:

Kurose S et al, Sci Rep 10:19129, 2020.

Reply: Thank you for the information. We have checked the contents and found them to be very important for our research. I have added it to the introduction and discussion.

Revised: Page 5, Line 82-83 <Introduction>

Revised: Page 23, Line 327-333 <Discussion>

---

## [Decision Letter · Decision Letter 1]

11 May 2021

PONE-D-20-34154R1

An association analysis between hypertension, dementia, and depression and the phases of pre-sarcopenia to sarcopenia: A cross-sectional analysis

PLOS ONE

Dear Dr. Endo,

Thank you for submitting your manuscript to PLOS ONE. After careful consideration, we feel that it has merit but does not fully meet PLOS ONE’s publication criteria as it currently stands. Therefore, we invite you to submit a revised version of the manuscript that addresses the points raised during the review process.

The manuscript has been improved; however, several revisions are still necessary in the present form. See the Reviewer #3's comments and respond them appropriately.

We look forward to receiving your revised manuscript.

Kind regards,

Masaki Mogi

Academic Editor

PLOS ONE

Reviewers' comments:

Reviewer's Responses to Questions

**Comments to the Author**

1. If the authors have adequately addressed your comments raised in a previous round of review and you feel that this manuscript is now acceptable for publication, you may indicate that here to bypass the “Comments to the Author” section, enter your conflict of interest statement in the “Confidential to Editor” section, and submit your "Accept" recommendation.

Reviewer #1: All comments have been addressed

Reviewer #2: All comments have been addressed

Reviewer #3: (No Response)

2. Is the manuscript technically sound, and do the data support the conclusions?

Reviewer #1: Yes

Reviewer #2: Yes

Reviewer #3: Partly

3. Has the statistical analysis been performed appropriately and rigorously? 

Reviewer #1: Yes

Reviewer #2: Yes

Reviewer #3: No

4. Have the authors made all data underlying the findings in their manuscript fully available?

Reviewer #1: Yes

Reviewer #2: Yes

Reviewer #3: Yes

5. Is the manuscript presented in an intelligible fashion and written in standard English?

Reviewer #1: Yes

Reviewer #2: Yes

Reviewer #3: Yes

6. Review Comments to the Author

Reviewer #1: I would like to thank the authors for their efforts and time. They have addressed my concerns properly. I have no further questions.

Reviewer #2: I recognized that authors addressed upon reviewers' points appropriately, and improved the work accordingly. I sincerely appreciate for the authors' effort.

Reviewer #3: The revised version has been partially improved. However, there are several major concerns with the manuscript as written that must be addressed.

Major comments:

1. The expression “shifting phase” is inappropriate in this study. Because the authors have not examined the predictors of subject who have shifted from non-sarcopenia to pre-sarcopenia. This study is a cross-sectional study, and its influence on the shifting phase is unclear.

2. SARC-F is not required for sarcopenia diagnosis using AWGS 2019, but can be diagnosed by grip strength, walking speed, and SMI. Therefore, the authors should address a clear reason for using AWGS2014.

3. The comparison of the three groups cannot be mentioned by only the Kruskal-Wallis test (p15, 243-248 line). Multiple comparison analysis is required to confirm these results.

Minor comments:

1. In Table 1, the interquartile range should be indicated by the 1st quartile – 3rd quartile. In addition, the authors should add the items such as pre-sarcopenia, sarcopenia etc in the column.

2. I could not find the S1 and S2 tables in this revised version.

3. The median of age was 74.9 years in this manuscript. However, the mean age of the first manuscript was also 74.9 years. Is it correct?

7. PLOS authors have the option to publish the peer review history of their article (what does this mean?). If published, this will include your full peer review and any attached files.

Reviewer #1: No

Reviewer #2: No

Reviewer #3: No

---

## [Author Response · Author response to Decision Letter 1]

19 May 2021

Response to reviewer 3

Reviewer 3 

Major comments:

1. The expression “shifting phase” is inappropriate in this study. Because the authors have not examined the predictors of subject who have shifted from non-sarcopenia to pre-sarcopenia. This study is a cross-sectional study, and its influence on the shifting phase is unclear.

Reply: 

　I changed the word to shifting instead of transition as pointed out by another reviewer.

You are right, the word shifting was inappropriate. I have revised it as follows.

Revised: page, line, 

　Page 7, Line 99-103

The purpose of this study is to examine the variables that influence the progressing stages of sarcopenia that impact robust, pre-sarcopenia, and sarcopenia in the residents of Oki-island. We hypothesized that depression and dementia have a positive effect, while hypertension has a negative effect, on the progression stages of sarcopenia.

　Page 21, Line 315-318

We found multiple factors associated with the progression from robust to pre-sarcopenia and sarcopenia. In the progression of stages from robust to pre-sarcopenia, in addition to age, depression and hypertension were significantly associated with the participants’ status.

2. SARC-F is not required for sarcopenia diagnosis using AWGS 2019, but can be diagnosed by grip strength, walking speed, and SMI. Therefore, the authors should address a clear reason for using AWGS2014.

Reply: 

As you pointed out, the SARC-F is not required for picking up sarcopenia because this study measured all participants. However, in this study, the speed of 5m walking was measured, so the speed of 6m walking is unknown. 6m walking speed should be measured by AWGS 2019 standards. Hence, we could not classify them according to the AWGS2019 criteria. The sentence about SARC-F has been removed from Limitation. I appreciate your pointing out our error.

Revised: page 25, Line 374-377

Third, this study was conducted in 2016, and the speed of the 5 m gait was measured according to the AWGS 2014 criteria and not the AWGS 2019 criteria using the 6 m gait. Our future task is to a conduct a sarcopenia study using the AWGS 2019 criteria.

3. The comparison of the three groups cannot be mentioned by only the Kruskal-Wallis test (p15, 243-248 line). Multiple comparison analysis is required to confirm these results.

Reply: 

In this part, we showed the trend of the three groups' characteristics. Then, we performed multivariate ordered logit regression analysis including the related factors and identified the significantly related factors (Table 2). Since the aim was achieved, we did not perform that multiple comparison analysis.

Minor comments:

1. In Table 1, the interquartile range should be indicated by the 1st quartile – 3rd quartile. In addition, the authors should add the items such as pre-sarcopenia, sarcopenia etc in the column.

Reply: 

I rewrote Table1 as you instructed. I had not written the items, robust, pre-sarcopenia, and sarcopenia; thus, I added them. Thank you for pointing this out.

2. I could not find the S1 and S2 tables in this revised version.

Reply: 

In accordance with the PLOS one MANUSCRIPT BODY FORMATTING GUIDELINES, the S1 and S2 tables (supporting information) were uploaded in a separate file and were not included in the manuscript.

3. The median of age was 74.9 years in this manuscript. However, the mean age of the first manuscript was also 74.9 years. Is it correct?

Reply: 

I made an error: mean age was exactly 74.89 years old and median age was 75.0 years old. I have corrected the text.

---

## [Decision Letter · Decision Letter 2]

24 May 2021

An association analysis between hypertension, dementia, and depression and the phases of pre-sarcopenia to sarcopenia: A cross-sectional analysis

PONE-D-20-34154R2

Dear Dr. Endo,

We’re pleased to inform you that your manuscript has been judged scientifically suitable for publication and will be formally accepted for publication once it meets all outstanding technical requirements.

Kind regards,

Masaki Mogi

Academic Editor

PLOS ONE

Additional Editor Comments (optional):

Reviewers' comments:

Reviewer's Responses to Questions

**Comments to the Author**

1. If the authors have adequately addressed your comments raised in a previous round of review and you feel that this manuscript is now acceptable for publication, you may indicate that here to bypass the “Comments to the Author” section, enter your conflict of interest statement in the “Confidential to Editor” section, and submit your "Accept" recommendation.

Reviewer #3: All comments have been addressed

2. Is the manuscript technically sound, and do the data support the conclusions?

Reviewer #3: Yes

3. Has the statistical analysis been performed appropriately and rigorously? 

Reviewer #3: Yes

4. Have the authors made all data underlying the findings in their manuscript fully available?

Reviewer #3: Yes

5. Is the manuscript presented in an intelligible fashion and written in standard English?

Reviewer #3: Yes

6. Review Comments to the Author

Reviewer #3: The revised version has been sufficiently addressed my concerns.

I would like to appreciate for the authors efforts.

7. PLOS authors have the option to publish the peer review history of their article (what does this mean?). If published, this will include your full peer review and any attached files.

Reviewer #3: No

---

## [Editor Report · Acceptance letter]

13 Jul 2021

PONE-D-20-34154R2 

An association analysis between hypertension, dementia, and depression and the phases of pre-sarcopenia to sarcopenia: A cross-sectional analysis 

Dear Dr. Endo:

I'm pleased to inform you that your manuscript has been deemed suitable for publication in PLOS ONE. Congratulations! Your manuscript is now with our production department. 

Kind regards, 

on behalf of

Dr. Masaki Mogi 

Academic Editor

PLOS ONE